# Trustworthy Multimodal Regression
# with Mixture of Normal-inverse Gamma Distributions

**Huan Ma**[*]
College of Intelligence and Computing
Tianjin University
Tianjin, China

**Zongbo Han**[*]
College of Intelligence and Computing
Tianjin University
Tianjin, China

**Changqing Zhang**[†]
College of Intelligence and Computing
Tianjin University
Tianjin, China

**Huazhu Fu**
Institute of High Performance Computing
A*STAR
Singapore

**Joey Tianyi Zhou**
Institute of High Performance Computing
A*STAR
Singapore

**Qinghua Hu**
College of Intelligence and Computing
Tianjin University
Tianjin, China

## Abstract

Multimodal regression is a fundamental task, which integrates the information from different sources to improve the performance of follow-up applications. However, existing methods mainly focus on improving the performance and often ignore the confidence of prediction for diverse situations. In this study, we are devoted to trustworthy multimodal regression which is critical in cost-sensitive domains. To this end, we introduce a novel Mixture of Normal-Inverse Gamma distributions (MoNIG) algorithm, which efficiently estimates uncertainty in principle for adaptive integration of different modalities and produces a trustworthy regression result. Our model can be dynamically aware of uncertainty for each modality, and also robust for corrupted modalities. Furthermore, the proposed MoNIG ensures explicitly representation of (modality-specific/global) epistemic and aleatoric uncertainties, respectively. Experimental results on both synthetic and different real-world data demonstrate the effectiveness and trustworthiness of our method on various multimodal regression tasks (e.g., temperature prediction for superconductivity, relative location prediction for CT slices, and multimodal sentiment analysis[3]).

## 1 Introduction

There are plenty of multimodal data involved in the real world, and we experience the world from different modalities [1]. For example, the autonomous driving systems are usually equipped with multiple sensors to collect information from different perspectives [2]. In medical diagnosis [3], multimodal data usually come from different types of examinations, typically including a variety of clinical data. Effectively exploiting the information of different sources to improve learning performance is a long-standing and challenging goal in machine learning.

---

[*]Equal contribution.
[†]Corresponding author. Please correspond to: zhangchangqing@tju.edu.cn.
[3]Code: https://github.com/MaHuanAAA/MoNIG.

35th Conference on Neural Information Processing Systems (NeurIPS 2021).

Most multimodal regression methods [4, 5, 6, 7] usually focus on improving the regression performance by exploiting the complementary information among multiple modalities. Despite effectiveness, it is quite risky for these methods to be deployed in cost-sensitive applications due to the lack of reliability and interpretability. One underlying deficiency is that traditional models usually assume the quality of each modality is basically stable, which limits them to produce reliable prediction, especially when some modalities are noisy or even corrupted [8, 9]. Moreover, existing models only output (over-confident) predictions [10, 11], which can not well support safe decision and might be disastrous for safety-critical applications.

Uncertainty estimation provides a way for trustworthy prediction [12, 13]. The decisions made by models without uncertainty estimation are untrustworthy because they are prone to be affected by noises or limited training data. Therefore, it is highly desirable to characterize uncertainty in the learning for AI-based systems. More specifically, when a model is given an input that has never been seen or is severely contaminated, it should be able to express "I don't know". The untrustworthy models are vulnerable to be attacked and it may also lead to wrong decisions, where the cost is often unbearable in critical domains [14].

Basically, it can endow the model with trustworthiness by dynamically modeling uncertainty. Therefore, we propose a novel algorithm that conducts multimodal regression in a trustworthy manner. Specifically, our proposed algorithm is a unified framework modeling uncertainty under a fully probabilistic framework. Our model integrates multiple modalities by introducing a Mixture of Normal-inverse Gamma distributions (MoNIG), which hierarchically characterizes the uncertainties and accordingly promotes both regression accuracy and trustworthiness. In summary, the contributions of this work include: (1) We propose a novel trustworthy multimodal regression algorithm. Our method effectively fuses multiple modalities under the evidential regression framework equipped with modality-specific uncertainty and global uncertainty. (2) To integrate different modalities, a novel MoNIG is designed to be dynamically aware of modality-specific noise/corruption with the estimated uncertainty, which promisingly supports trustworthy decision making and also significantly promotes robustness as well. (3) We conduct extensive experiments on both synthetic and real-application data, which validate the effectiveness, robustness, and reliability of the proposed model on different multimodal regression tasks (*e.g.*, critical temperature prediction for superconductivity [15], relative location of CT slices, and human multimodal sentiment analysis).

## 2   Related Work

### 2.1   Uncertainty Estimation

Quantifying the uncertainty of machine learning models has received extensive attention [16, 17], especially when the systems are deployed in safety-critical domains, such as autonomous vehicle control [18] and medical diagnosis [3]. **Bayesian neural networks** [19, 20] model the uncertainty by placing a distribution over model parameters and marginalizing these parameters to form a predictive distribution. Due to the huge parameter space of modern neural networks, Bayesian neural networks are highly non-convex and difficult in inference. To tackle this problem, [21] extends Variational Dropout [22] to the case when dropout rates are unbounded, and proposes a way to reduce the variance of the gradient estimator. A more scalable alternative way is MC Dropout [23], which is simple to implement and has been successfully applied to downstream tasks [14, 24]. **Deep ensembles** [25] have shown strong power in both classification accuracy and uncertainty estimation. It is observed that deep ensembles consistently outperform Bayesian neural networks that are trained using variational inference [10]. However, the memory and computational cost are quite high. For this issue, different deep sub-networks with shared parameters are trained for integration [26]. **Deterministic uncertainty methods** are designed to directly output the uncertainty and alleviate the overconfidence. Built upon RBF networks, [27] is able to identify the out-of-distribution samples. [28] introduces a new target criterion for model confidence, known as True Class Probability (TCP), to ensure the low confidence for the failure predictions. The recent approach employs the focal loss to calibrate the deep neural networks [29]. [30] places Dirichlet priors over discrete classification predictions and regularizes divergence to a well-defined prior. Our model is inspired by deep evidential regression [31] which is designed for single-modal data.

## 2.2 Multimodal Learning

Multimodal machine learning aims to build models that can jointly exploit information from multiple modalities [1, 32]. Existing multimodal learning methods have achieved significant progress by integrating different modalities at different stages, namely early, intermediate and late fusion [1, 33]. The early fusion methods usually integrate the original data or preprocessed features by simply concatenation [34, 35]. The intermediate fusion provides a more flexible strategy to exploit multimodal data for diverse practical applications [4, 36, 37, 38, 39, 40, 41]. In late fusion, each modality is utilized to train a separate model and the final output is obtained by combining the predictions of these multiple models [5, 7, 37].

Although these multimodal learning approaches exploit the complementary information of multiple modalities from different perspectives, basically they are weak in modeling uncertainty which is important for trustworthy prediction especially when they are deployed in safety-critical domains.

## 3 Background

Regression has been widely used across a wide spectrum of applications. Given a dataset $\mathcal{D} = \{\mathbf{x}_i, y_i\}_{i=1}^N$, one principled way for regression is from a maximum likelihood perspective, where the likelihood for parameters $\mathbf{w}$ given the observed data $\mathcal{D}$ is as follows:

$$p(\mathbf{y} \mid \mathcal{D}_x, \mathbf{w}) = \prod_{i=1}^N p(y_i \mid \mathbf{w}, \mathbf{x}_i). \tag{1}$$

In practice, a widely used maximum likelihood estimation is based on Gaussian distribution, which assumes that the target $y$ is drawn from a Gaussian distribution. Then we have the following expression of likelihood function:

$$p(\mathbf{y} \mid \mathcal{D}_x, \mathbf{w}, \sigma^2) = \prod_{i=1}^N \mathcal{N}\left(y_i \mid f\left(\mathbf{x}_i, \mathbf{w}\right), \sigma^2\right). \tag{2}$$

The mean and variance indicate the prediction and corresponding predictive uncertainty, while the predictive uncertainty consists of two parts [42]: epistemic uncertainty (EU) and aleatoric uncertainty (AU). To learn both the aleatoric and epistemic uncertainties explicitly, the mean and variance are assumed to be drawn from Gaussian and Inverse-Gamma distributions [31], respectively. Then the Normal Inverse-Gamma (NIG) distribution with parameters $\boldsymbol{\tau} = (\delta, \gamma, \alpha, \beta)$ can be considered as a higher-order conjugate prior of the Gaussian distribution parameterized with $\boldsymbol{\theta} = \left(\mu_m, \sigma^2\right)$:

$$y_i \sim \mathcal{N}\left(\mu, \sigma^2\right), \quad \mu \sim \mathcal{N}\left(\delta, \sigma^2\gamma^{-1}\right), \quad \sigma^2 \sim \Gamma^{-1}(\alpha, \beta), \tag{3}$$

where $\Gamma(\cdot)$ is the gamma function. In this case, the distribution of $y$ takes the form of a NIG distribution NIG$(\delta, \gamma, \alpha, \beta)$:

$$p(\underbrace{\mu, \sigma^2}_{\boldsymbol{\theta}} \mid \underbrace{\delta, \gamma, \alpha, \beta}_{\boldsymbol{\tau}}) = \frac{\beta^\alpha}{\Gamma(\alpha)} \frac{\sqrt{\gamma}}{\sigma\sqrt{2\pi}} \left(\frac{1}{\sigma^2}\right)^{\alpha+1} \exp\left\{-\frac{2\beta + \gamma(\delta - \mu)^2}{2\sigma^2}\right\}, \tag{4}$$

where $\delta \in \mathrm{R}, \gamma > 0, \alpha > 1$, and $\beta > 0$.

During the training stage, the following loss is induced to minimize the negative log likelihood loss:

$$\mathcal{L}^{NLL}(\mathbf{w}) = \frac{1}{2}\log\left(\frac{\pi}{\gamma}\right) - \alpha\log(\Omega) + \left(\alpha + \frac{1}{2}\right)\log\left((y - \delta)^2\gamma + \Omega\right) + \log\Psi, \tag{5}$$

where $\Omega = 2\beta(1 + \gamma)$ and $\Psi = \left(\frac{\Gamma(\alpha)}{\Gamma(\alpha + \frac{1}{2})}\right)$.

The total loss, $\mathcal{L}(\mathbf{w})$, consists of two terms for maximizing the likelihood function and regularizing evidence:

$$\mathcal{L}(\mathbf{w}) = \mathcal{L}^{NLL}(\mathbf{w}) + \lambda\mathcal{L}^R(\mathbf{w}), \tag{6}$$

where $\mathcal{L}^{\mathrm{R}}(\mathbf{w})$ is the penalty for incorrect evidence (more details are shown in the supplement), and the coefficient $\lambda > 0$ balances these two loss terms.

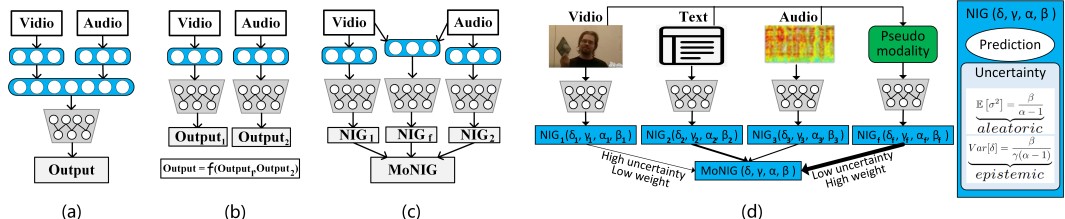

Figure 1: Strategies for multimodal regression. Feature fusion (a), decision fusion (b), the proposed MoNIG (c), and an instance of MoNIG on the human multimodal sentiment analysis task (d).

For multimodal regression [5, 7, 6], some approaches have achieved impressive performance. However, existing multimodal regression algorithms mainly focus on improving the accuracy and provide a deterministic prediction without the information of uncertainty, making these models limited for trustworthy decision making. For this issue, we develop a novel algorithm which can capture the modality-specific uncertainty and accordingly induces a reliable overall uncertainty for the final output. Beyond trustworthy decision, our approach automatically alleviates the impact from heavily noisy or corrupted modality. Compared with the intermediate fusion [4, 43, 44], our method can effectively distinguish which modality is noisy or corrupted for different samples, and accordingly can take the modality-specific uncertainty into account for robust integration.

## 4 Trustworthy Multimodal Regression

In this section, we introduce the proposed algorithm fusing multiple modalities at both feature and predictive distribution levels. Specifically, we train deep neural networks to model the hyperparameters of the higher-order evidential distributions of multiple modalities (and the pseudo modality) and then merge these predicted distributions into one by the mixture of NIGs (MoNIG). In the following subsections, we first provide discussion about existing fusion strategies and ours in multimodal regression. Then we introduce MoNIG to integrate NIG distributions of different modalities in principle. Finally, we elaborate the overall pipeline of our model in training and define the epistemic uncertainty and aleatoric uncertainty respectively.

### 4.1 Overview of Fusion Strategy

Consider the following multimodal regression problem: given a dataset $\mathcal{D} = \left\{ \{\mathbf{x}_i^m\}_{m=1}^M, y_i \right\}$, where $\mathbf{x}_i^m$ is the input feature vector of the $m$-th modality of the $i$-th sample, $y_i$ is the corresponding target, and $M$ is the total number of modalities, the intuitive goal is to learn a function $f_m$ for each modality reaching the following target: $\min \sum_{m=1}^M \left( f_m(\mathbf{x}_m, \mathbf{w}_m) - y \right)^2$. While to model the uncertainty, we assume that the observed target $y$ is drawn from a Gaussian distribution, i.e., $y \sim \mathcal{N}(\mu_m, \sigma_m^2)$. Meanwhile, inspired by [31], as shown in Eq. 3, we assume that the mean and variance are drawn from Gaussian and Inverse-Gamma distributions, respectively.

Considering multimodal fusion, there are typically two representative strategies, i.e., feature fusion and decision fusion. Feature fusion (Fig. 1(a)) jointly learns unified representations and based on which conducts regression. Although simple, the limitation is also obvious. If some modalities of some samples are noisy or corrupted, the algorithms based on this strategy may fail. In other words, these methods do not take the modality-specific uncertainty into account and can not be adaptively aware of quality variation of different modalities for different samples. The other representative strategy is known as decision fusion (Fig. 1(b)) which trains one regression model for each modality, and the final prediction is regarded as a function of the predictions of all modalities. Although flexible, it is also risky for the cases that each modality is insufficient to make trustworthy decision due to partial observation.

We propose MoNIG, which can elegantly address the above issues in a unified framework. Firstly, we explicitly represent both modality-specific and global uncertainties to endow our model with the ability of adaption for dynamical modality variation. The modality-specific uncertainty can dynamically guide the integration for different samples and the global uncertainty represents the

uncertainty of the final prediction. Secondly, we introduce the feature level fusion producing a pseudo modality to make full use of the complementary information and, accordingly generate an enhanced branch to supplement the decision fusion (Fig 1(c)).

## 4.2 Mixture of Normal-inverse Gamma Distributions

In this section, we focus on fusing multiple NIG distributions from original modalities and the generated pseudo modality. The key challenge is how to reasonably integrate multiple NIG distributions into a uniform NIG. The widely used product of experts (PoE) [45] is not suitable for our case since there is no guarantee that the product of multiple NIGs is still a NIG. Moreover, integration with PoE tends to be affected by noisy modalities [46]. For these issues, another natural way is integrating a set of NIGs with the following additive strategy:

$$y \sim \sum_{m=1}^{M} \frac{1}{M} NIG(\delta_m, \gamma_m, \alpha_m, \beta_m). \tag{7}$$

Although simple in form, there are two main limitations for Eq. 7. First, simply averaging over all NIGs does not take the uncertainties of different NIGs into account, and thus it may be seriously affected by noisy modalities. Second, it is intractable to infer the parameters for the fused NIG distribution in practice since there is no closed-form solution. Therefore, we introduce the NIG summation operation [47] to approximately solve this problem. Specifically, the NIG summation operator defines a novel operation to ensure a new NIG distribution after the fusion of two NIG distributions.

**Definition 4.1.** *(**Summation of NIG distributions**) Given two NIG distributions, i.e., $NIG(\delta_1, \gamma_1, \alpha_1, \beta_1)$ and $NIG(\delta_2, \gamma_2, \alpha_2, \beta_2)$, the definition of the summation of these two NIG distributions is*

$$NIG(\delta, \gamma, \alpha, \beta) \triangleq NIG(\delta_1, \gamma_1, \alpha_1, \beta_1) \oplus NIG(\delta_2, \gamma_2, \alpha_2, \beta_2), \tag{8}$$

*where*

$$\delta = (\gamma_1 + \gamma_2)^{-1}(\gamma_1\delta_1 + \gamma_2\delta_2), \qquad \alpha = \alpha_1 + \alpha_2 + \frac{1}{2},$$
$$\gamma = \gamma_1 + \gamma_2, \quad \beta = \beta_1 + \beta_2 + \frac{1}{2}\gamma_1(\delta_1 - \delta)^2 + \frac{1}{2}\gamma_2(\delta_2 - \delta)^2. \tag{9}$$

Therefore, fusing NIG distributions in this way endows our model with several promising properties (shown in Pro. 4.1) which allow us to use it for trustworthy multimodal regression based on uncertainty. Then we substitute Eq. 7 with the following operation:

$$NIG(\delta, \gamma, \alpha, \beta) = NIG(\delta_1, \gamma_1, \alpha_1, \beta_1) \oplus NIG(\delta_2, \gamma_2, \alpha_2, \beta_2) \oplus \cdots \oplus NIG(\delta_M, \gamma_M, \alpha_M, \beta_M), \tag{10}$$

where $\oplus$ represents the summation operation of two NIG distributions.

The NIG summation can reasonably make use of modalities with different qualities. Specifically, the parameter $\gamma_m$ indicates the confidence of a NIG distribution for the mean $\delta_m$ [48]. As shown in Def. 4.1, if one modality is more confident with its prediction then it will contribute more to the final prediction. Moreover, $\beta$ directly reflects both aleatoric uncertainty and epistemic uncertainty (Eq. 12) which consists of two parts, i.e., the sum of $\beta_1$ and $\beta_2$ from multiple modalities and the variance between the final prediction and that of every single modality. Intuitively, the final uncertainty is determined jointly by the modality-specific uncertainty and the prediction deviation among different modalities.

**Proposition 4.1.** *The summation operation in Definition 4.1 of NIG distributions has the following properties:*
*1. **Commutativity**:*

$$NIG\left(\delta_1, \gamma_1, \alpha_1, \beta_1\right) \oplus NIG\left(\delta_2, \gamma_2, \alpha_2, \beta_2\right) = NIG\left(\delta_2, \gamma_2, \alpha_2, \beta_2\right) \oplus NIG\left(\delta_1, \gamma_1, \alpha_1, \beta_1\right).$$

*2. **Associativity**:*

$$NIG\left(\delta_1, \gamma_1, \alpha_1, \beta_1\right) \oplus NIG\left(\delta_2, \gamma_2, \alpha_2, \beta_2\right) \oplus NIG\left(\delta_3, \gamma_3, \alpha_3, \beta_3\right)$$
$$= NIG\left(\delta_1, \gamma_1, \alpha_1, \beta_1\right) \oplus [NIG\left(\delta_2, \gamma_2, \alpha_2, \beta_2\right) \oplus NIG\left(\delta_3, \gamma_3, \alpha_3, \beta_3\right)].$$

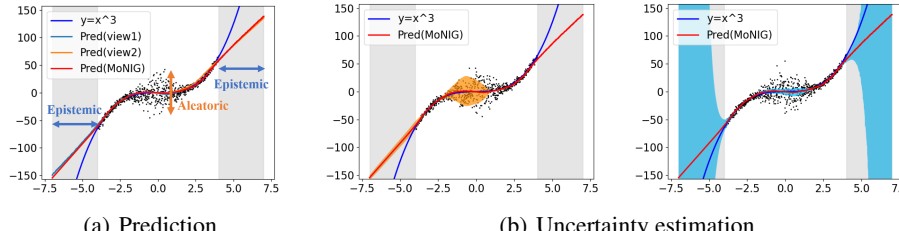

(a) Prediction           (b) Uncertainty estimation

Figure 2: (a) Regressed curves for each modality and the fused ones (with MoNIG), where gray shadow areas indicate there is no training data available; (b) the left and right subfigures demonstrate the estimated aleatoric and epistemic uncertainties, respectively.

The above two properties can be easily proved (refer to the supplement). Based on the summation operation, our multimodal regression algorithm has the following advantages. **(1) Flexibility**: according to Def. 4.1 and Pro. 4.1, we can fuse an arbitrary number of NIG distributions conveniently. **(2) Explainability**: we can explain the detailed fusion process for multiple NIG distributions and observe the belief degree of each NIG distribution according to Eq. 9. **(3) Trustworthiness**: using Def. 4.1 allows us to fuse multiple NIG distributions into a new NIG distribution, which is critical to explicitly provide both epistemic uncertainty and aleatoric uncertainty for trustworthy decision making. **(4) Optimizability**: compared with struggling to seek a closed-form solution and conducting complex optimization, it is much more efficient to optimize using Def. 4.1 for multiple NIGs fusion.

### 4.3 Overall Learning Framework

Inspired by multi-task learning, we define the final loss function as the sum of losses of multiple modalities (including the pseudo modality) and the predictive-level fused distribution:

$$\mathcal{L}(\mathbf{w}) = \sum_{m=1}^{M} \mathcal{L}_m(\mathbf{w}) + \mathcal{L}_P(\mathbf{w}) + \mathcal{L}_{MoNIG}(\mathbf{w}), \tag{11}$$

where $\mathcal{L}_m(\mathbf{w})$ is obtained according to Eq. 6. Specifically, $\mathcal{L}_P(\mathbf{w})$ is the pseudo modality loss defined as $\mathcal{L}_P(\mathbf{w}) = \mathcal{L}_P^{NLL}(\mathbf{w}) + \lambda \mathcal{L}_P^{R}(\mathbf{w})$, and $\mathcal{L}_{MoNIG}(\mathbf{w})$ is the fused distribution loss defined as $\mathcal{L}_{MoNIG}(\mathbf{w}) = \mathcal{L}_{MoNIG}^{NLL}(\mathbf{w}) + \lambda \mathcal{L}_{MoNIG}^{R}(\mathbf{w})$. More detailed derivation for the overall loss is in the supplement. The proposed method is a general uncertainty-aware fusion module so that the pseudo modality may be obtained in different ways (e.g. features concatenation or concatenation after representation learning).

Given the MoNIG distribution, the aleatoric and epistemic uncertainties are defined as:

$$\underbrace{\mathbb{E}\left[\sigma^2\right] = \frac{\beta}{\alpha - 1}}_{aleatoric}, \qquad \underbrace{Var[\delta] = \frac{\beta}{\gamma(\alpha - 1)}}_{epistemic}. \tag{12}$$

For clarification, we provide the following remarks: (1) The objective is a unified learnable framework, and thus the branches of original modalities, pseudo modality, and the fused global branch can be improved with each other with the multi-task learning strategy; (2) The final output is explainable according to the local (modality-specific) and global (fused) uncertainties, and the aleatoric and epistemic uncertainties as well.

## 5 Experiments

To demonstrate the effectiveness of our model, we conduct experiments on synthetic and real-world data including physical (Superconductivity[4]), medical (CT slices[5]), and multimodal sentiment

---

[4]https://archive.ics.uci.edu/ml/datasets/Superconductivity
[5]https://archive.ics.uci.edu/ml/datasets/CT+slices

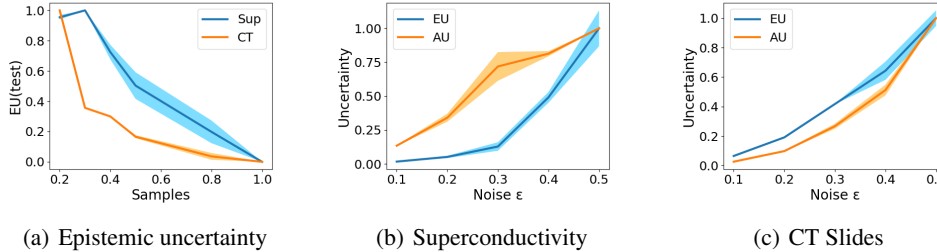

| (a) Epistemic uncertainty | (b) Superconductivity | (c) CT Slides |

Figure 3: Epistemic uncertainty with different number of training samples (a), and relationship between the uncertainty and noise degree (b)-(c).

analysis tasks[6]. Furthermore, we also conduct experiments to validate the effectiveness of uncertainty estimation in a variety of conditions. Finally, the ablation study is conducted further assessing the effectiveness of the approach in terms of uncertainty quantification and investigating whether the promising performance is due to the novel strategy.

### 5.1 Experiment on Synthetic Data

To intuitively illustrate the properties of our model, firstly we conduct experiments on synthetic data. Following [31, 49], we conduct sampling with $y = x^3 + \epsilon$, where $\epsilon \sim \mathcal{N}(0,3)$ for $x \in [-4, -2] \bigcup [2, 4]$. While for the sampling data points $x \in (2, -2)$ we add stronger noise on input, where $\epsilon \sim \mathcal{N}(0, \sigma)$ with $\sigma = 3 + 0.8 \times (2 - |x|)$, and the $m$-th modality input $x_m = x + \epsilon_x$, $\epsilon_x \sim \mathcal{N}(0, 0.01)$. The visualization and corresponding analysis could be found in Fig. 2, where the left and right subfigures demonstrate the prediction ability and estimated aleatoric/epistemic uncertainties, respectively. According to Fig. 2(a), it is observed that all curves fit well to the ground-truth curve given enough training data. From the left subfigure of Fig. 2(b), we find that the estimated aleatoric uncertainty (**AU**) ($x \in [-2, 2]$) is much higher compared with that of $x \in [-4, -2] \bigcup [2, 4]$, which is consistent with the basic assumption. For the right subfigure, it demonstrates that the epistemic uncertainty (**EU**) is much higher when there is no training data, validating that our model can well characterize the epistemic uncertainty.

### 5.2 Temperature Prediction for Superconductivity

**Superconductivity** contains 21263 samples, and all samples are described by two modalities. The first one contains 81 features of its experimental properties, and the second contains the chemical formula extracted into 86 features. The goal here is to predict the critical temperature (values are in the range [0, 185]) based on these two modalities. In our experiment, we use 10633, 4000, and 6600 samples as training, validation, and test data, respectively. We compare our model with existing approaches including Gaussian (**GS**) in Eq. 2, Evidential (**EVD**) [31], where "IF" and "DF" represent different fusion strategies, i.e, "IF" and "DF" denoting intermediate level (concatenating the features from hidden layer) and data level (concatenating the original features) fusion, respectively. "Pseudo"

Table 1: Multimodal regression (RMSE $\downarrow$).

| Mod1 | Mod2 | GS (IF/DF) | EVD (IF/DF) | Ours (Pseudo) |
| --- | --- | --- | --- | --- |
| 12.93 | 12.82 | 13.93/13.78 | 14.18/12.79 | **12.19 (11.70)** |

Table 2: OOD (out-of-distribution) detection (AU-ROC $\uparrow$) using the estimated uncertainty.

| Modality | Noise $\epsilon$ | GS | EVD | | Ours | |
| --- | --- | --- | --- | --- | --- | --- |
| | | | AU | EU | AU | EU |
| Mod1 | 0.1 | 0.503 | 0.503 | 0.494 | **0.586** | **0.536** |
| | 0.5 | 0.511 | 0.486 | 0.424 | **0.782** | **0.551** |
| Mod2 | 0.1 | 0.503 | 0.507 | 0.486 | **0.839** | **0.577** |
| | 0.5 | 0.512 | 0.506 | 0.439 | **0.858** | **0.539** |
| RandMod | 0.1 | 0.502 | 0.502 | 0.488 | **0.699** | **0.549** |
| | 0.5 | 0.508 | 0.492 | 0.426 | **0.812** | **0.539** |
| AllMod | 0.1 | 0.507 | 0.509 | 0.484 | **0.838** | **0.584** |
| | 0.5 | 0.518 | 0.494 | 0.424 | **0.887** | **0.546** |

indicates that the pseudo modality (concatenating the features from hidden layer) is involved. The performance from our approach (with the Adam optimizer: $learning\_rate = 1e - 3$ for 400 iterations) is superior to the compared methods, which is with the lowest Root Mean Squared Error (RMSE). We also impose different degrees of noise in different conditions to half of the test samples as out-of-distribution (OOD) samples, and try to distinguish them with uncertainty. "AU" and "EU"

---

[6]http://immortal.multicomp.cs.cmu.edu

indicate the uncertainties used to distinguish OOD samples. According to Table 2, our model achieves clearly better performance in terms of AUROC on different noise conditions (a higher value indicates a better uncertainty estimation).

## 5.3 Relative Location Prediction for CT Slices

**CT Slices** are retrieved from a set of 53500 CT images from 74 different patients. Each CT slice is described by two modalities. The first modality describes the location of bone structures in the image, containing 240 attributes. The second modality describes the location of air inclusions inside of the body, containing 144 attributes. The goal is to predict the relative location of the image on the axial axis. The target values are in the range [0, 180], where 0 denotes the top of the head and 180 the soles of the feet. We divide the dataset into 26750/10000/16750 samples as training, validation, and test data, respectively. The other settings are the same as those on the Superconductivity dataset. We also test the ability of OOD detection and the results are reported in Table 4. It is clear that our model achieves much better performance than comparisons.

Table 3: Multimodal regression (RMSE ↓).

| Mod1 | Mod2 | GS (IF/DF) | EVD (IF/DF) | Ours (Pseudo) |
|---|---|---|---|---|
| 1.67 | 4.49 | 1.64/1.70 | 2.97/3.27 | **0.91 (0.79)** |

Table 4: OOD (out-of-distribution) detection (AU-ROC ↑) using the estimated uncertainty.

| Modality | Noise $\epsilon$ | GS | EVD | | Ours | |
|---|---|---|---|---|---|---|
| | | | AU | EU | AU | EU |
| Mod1 | 0.1 | 0.504 | 0.504 | 0.502 | **0.532** | **0.543** |
| | 0.5 | 0.582 | 0.638 | 0.610 | **0.771** | **0.806** |
| Mod2 | 0.1 | 0.503 | 0.503 | 0.500 | **0.519** | **0.511** |
| | 0.5 | 0.529 | 0.600 | 0.584 | **0.870** | **0.866** |
| RandMod | 0.1 | 0.503 | 0.504 | 0.505 | **0.569** | **0.582** |
| | 0.5 | 0.570 | 0.619 | 0.601 | **0.835** | **0.852** |
| AllMod | 0.1 | 0.507 | 0.509 | 0.508 | **0.600** | **0.615** |
| | 0.5 | 0.618 | 0.672 | 0.631 | **0.808** | **0.839** |

## 5.4 Human Multimodal Sentiment Analysis

**CMU-MOSI & MOSEI.** CMU-MOSI [50] is a multimodal sentiment analysis dataset consisting of 2,199 short monologue video clips, while MOSEI [51] is a sentiment and emotion analysis dataset consisting of 23,454 movie review video clips taken from YouTube. Each task consists of a word-aligned and an unaligned version. For both versions, the multimodal features are extracted from the textual [52], visual and acoustic [53] modalities.

Table 5: Results on CMU-MOSI.

| Metric | Acc$_7$↑ | Acc$_2$↑ | F1↑ | MAE↓ | Corr↑ |
|---|---|---|---|---|---|
| (Word Aligned) CMU-MOSI Sentiment | | | | | |
| EF-LSTM | 33.7 | 75.3 | 75.2 | 1.023 | 0.608 |
| LF-LSTM | **35.3**[2] | 76.8 | 76.7 | 1.015 | 0.625 |
| RAVEN [54] | 33.2 | 78.0 | 76.6 | **0.915**[2] | **0.691**[1] |
| MCTN [55] | **35.6**[1] | 79.3 | 79.1 | **0.909**[1] | 0.676 |
| Gaussian | 32.9 | 78.4 | 78.4 | 0.982 | 0.657 |
| Evidential | 33.4 | 77.6 | 77.6 | 0.974 | 0.655 |
| MoNIG | 33.0 | **80.2**[2] | **80.4**[2] | 0.959 | 0.664 |
| MoNIG (pseudo) | 34.1 | **80.6**[1] | **80.6**[1] | 0.951 | **0.680**[2] |
| (Unaligned) CMU-MOSI Sentiment | | | | | |
| CTC + EF-LSTM [56] | 31.0 | 73.6 | 74.5 | 1.078 | 0.542 |
| LF-LSTM | 33.7 | 77.6 | 77.8 | 0.988 | 0.624 |
| CTC + MCTN [55] | 32.7 | 75.9 | 76.4 | 0.991 | 0.613 |
| CTC + RAVEN [54] | 31.7 | 72.7 | 73.1 | 1.076 | 0.544 |
| Gaussian | 32.4 | 77.6 | 77.5 | 1.005 | 0.634 |
| Evidential | 32.9 | 78.7 | 78.7 | 0.988 | 0.651 |
| MoNIG | **35.8**[1] | **79.3**[1] | **79.3**[1] | **0.972**[2] | **0.664**[2] |
| MoNIG (pseudo) | **34.7**[2] | **79.1**[2] | **79.1**[2] | **0.958**[1] | **0.669**[1] |

Table 6: Results on CMU-MOSEI.

| Metric | Acc$_7$↑ | Acc$_2$↑ | F1↑ | MAE↓ | Corr↑ |
|---|---|---|---|---|---|
| (Word Aligned) CMU-MOSEI Sentiment | | | | | |
| EF-LSTM | 47.4 | 78.2 | 77.9 | 0.642 | 0.616 |
| LF-LSTM | 48.8 | 80.6 | 80.6 | 0.619 | 0.659 |
| RAVEN [54] | **50.0**[2] | 79.1 | 79.5 | 0.614 | 0.662 |
| MCTN [55] | 49.6 | 79.8 | 80.6 | 0.609 | 0.670 |
| Gaussian | 49.3 | **81.6**[2] | **81.9**[2] | 0.613 | 0.677 |
| Evidential | 48.9 | 81.0 | 81.2 | 0.612 | 0.671 |
| MoNIG | **50.2**[1] | **81.8**[1] | **82.0**[1] | **0.602**[2] | **0.682**[2] |
| MoNIG (pseudo) | **50.0**[2] | 81.0 | 81.5 | **0.600**[1] | **0.688**[1] |
| (Unaligned) CMU-MOSEI Sentiment | | | | | |
| CTC + EF-LSTM [56] | 46.3 | 76.1 | 75.9 | 0.680 | 0.585 |
| LF-LSTM | 48.8 | 77.5 | 78.2 | 0.624 | 0.656 |
| CTC + RAVEN [54] | 45.5 | 75.4 | 75.7 | 0.664 | 0.599 |
| CTC + MCTN [55] | 48.2 | 79.3 | 79.7 | 0.631 | 0.645 |
| Gaussian | 48.8 | 81.0 | 81.3 | 0.618 | **0.676**[2] |
| Evidential | 49.2 | 81.3 | 81.7 | **0.608**[2] | **0.676**[2] |
| MoNIG | **50.7**[1] | **81.7**[1] | **81.9**[2] | **0.598**[1] | **0.693**[1] |
| MoNIG (pseudo) | **49.5**[2] | **81.7**[1] | **82.0**[1] | 0.612 | 0.673 |

For CMU-MOSEI dataset, similarly to existing work, there are 16326, 1871, and 4659 samples used as the training, validation, and test data, respectively. For CMU-MOSI dataset, we use 1284, 229, and 686 samples as training, validation, and test data, respectively. Similar to previous work [51, 55], we employ diverse metrics for evaluation: 7-class accuracy (Acc7), binary accuracy (Acc2), F1 score, mean absolute error (MAE), and the correlation (Corr) of the model's prediction with human. We directly concatenate the features extracted from the temporal convolutional layers of different networks corresponding to different modalities as a pseudo modality. Our method achieves competitive performance even compared with the state-of-the-art multimodal sentiment classification methods.

## 5.5 Uncertainty Estimation

| (a) $\epsilon = 0.1$ | (b) $\epsilon = 0.3$ | (c) $\epsilon = 0.5$ | (d) $\epsilon = 1.0$ |

Figure 4: Sensitivity in identifying noisy modality. We randomly select one from two modalities and add different degree of noise on it.

To validate the ability of epistemic estimation of our model in real data, we gradually increase the ratio of training samples from $20\%$ to $100\%$ of all training data. According to Fig. 3(a), we can find that the overall epistemic uncertainty declines steadily as the number of training samples increases.

We illustrate the relationship between the uncertainty and different degree of noise (Fig. 3(b)-(c)). It is observed that as the noise goes stronger, both aleatoric and epistemic uncertainties become larger, which implies that our model can be adaptively aware of possible noise in practical tasks. It should be pointed out that data noise (AU) will also significantly affect the EU under limited training data.

To investigate the sensitivity for noisy modality, we add different degrees of Gaussian noise (i.e., zero mean and varying variance $\epsilon$) to one of two modalities which is randomly selected. There are 500 samples that are associated with noisy modality 1 and noisy modality 2, respectively. As shown in Fig. 4, our algorithm can effectively distinguish which modality is noisy for different samples. Overall, the proposed method can capture global uncertainty (Fig. 3), and also has a very effective perception of modality-specific noise. Accordingly, the proposed algorithm can provide potential explanation for erroneous prediction.

## 5.6 Ablation Study

**Robustness to noise.** We add different degrees of Gaussian noise (i.e., zero mean and varying variance $\epsilon$) to one of two modalities which is randomly selected, then compare our method with EVD [31] (concatenating the original features). The results in Table 7 validate that our method can dexterously estimate the uncertainty for promising performance due to taking the modality-specific uncertainty into account for the integration, while it is difficult for the counterpart, i.e., EVD with concatenated features.

Table 7: Comparison between EVD [31] and ours (RMSE ↓).

| Dataset | Method | $\epsilon = 0.01$ | $\epsilon = 0.05$ | $\epsilon = 0.1$ |
|---------|--------|-------|-------|-------|
| Sup | EVD | 15.75 | 28.91 | 44.80 |
|     | Our | **13.95** | **17.38** | **21.82** |
| CT | EVD | 2.45 | 3.40 | 5.35 |
|    | Ours | **0.97** | **1.34** | **2.23** |

**Comparison with different decision fusion strategies.** To clarify which part contributes to the improvements, we compare different decision fusion strategies: average, weighted average with AU, and weighted average with EU. Since our method employs the modality-specific uncertainty to dynamically guide the integration for different samples, ours performs as the best as shown in Table 8.

Table 8: Comparison between different fusion strategies (RMSE ↓).

| Dataset | Average | Weighted (AU/EU) | Ours |
|---------|---------|------------------|------|
| Sup | 13.01 | 13.17/13.03 | **12.19** |
| CT | 2.87 | 5.71/1.11 | **0.91** |

**Effectiveness of uncertainty estimation.** To quantitatively investigate the uncertainty estimation, we define a rank-based criterion to compare ours with other methods, which directly measures the inconsistency between the estimated uncertainty and predictive error. The uncertainty-error inconsistency rate (UEIR)

Table 9: Quantitative evaluation for uncertainty (UEIR ↓).

| Method | AU (EVD/Ours) | EU (EVD/Ours) |
|--------|---------------|---------------|
| **UEIR (%)** | 13.73/**12.65** | 12.37/**11.70** |

is defined as UEIR $= \frac{\text{Num}_{inconst}}{\text{Num}_{all}} \times 100\%$, where $\text{Num}_{inconst}$ is the number of sample pairs that RMSE and uncertainty are in the opposite order (e.g., $\text{RMSE}_i > \text{RMSE}_j$ & $\text{uncertainty}_i < \text{uncertainty}_j$),

and Num$_{all}$ is the total number of pairs for test samples. A smaller value implies a better uncertainty estimation. More detailed description of Table 8 and Table 9 is shown in the supplementary material.

## 6 Conclusion

In this paper, we propose a novel trustworthy multimodal regression model, which elegantly characterizes both modality-specific and fused uncertainties and thus can provide trustworthiness for the final output. This is quite important for real-world (especially cost-sensitive) applications. The introduced pseudo modality further enhances the exploration of complementarity among multiple modalities producing more stable performance. The proposed model is a unified learnable framework, and thus can be efficiently optimized. Extensive experimental results on diverse applications and conditions validate that our model achieves impressive performance in terms of regression accuracy and also obtains reasonable uncertainty estimation to ensure trustworthy decision making. It is interesting to apply the proposed algorithm for more real-world applications in the future. Moreover, it is also important to extend the regression algorithm for trustworthy multimodal classification.

## Acknowledgements

This work was supported in part by the National Natural Science Foundation of China under Grant 61976151, 61732011, and the Natural Science Foundation of Tianjin of China under Grant 19JCY-BJC15200. We thank Alexander Amini (Massachusetts Institute of Technology), the author of the paper [31], for his generous help in explaining his work and providing useful reference materials.

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
