## A    Detailed derivation of the loss function

In this section, we elaborate on the training of neural networks to estimate the evidential distribution for each modality. From Bayesian probability theory, we marginalize over the likelihood parameters $\boldsymbol{\theta_m}$ to obtain the model evidence. Specifically, we should maximize the following likelihood function:

$$p(y|\boldsymbol{\tau_m}) = \frac{p(y|\boldsymbol{\theta_m}, \boldsymbol{\tau_m})p(\boldsymbol{\theta_m}|\boldsymbol{\tau_m})}{p(\boldsymbol{\theta_m}|y, \boldsymbol{\tau_m})} = \int_{\sigma_m^2=0}^{\infty} \int_{\mu_m=-\infty}^{\infty} p(y|\mu_m, \sigma_m^2)p(\mu_m, \sigma_m^2|\boldsymbol{\tau_m}) \, d\mu_m \, d\sigma_m^2,$$

(13)

where $\boldsymbol{\theta_m} = (\mu_m, \sigma_m^2)$ are the likelihood parameters and $\boldsymbol{\tau_m} = (\delta_m, \gamma_m, \alpha_m, \beta_m)$ are the evidential distribution parameters. Since it is difficult to compute the exact posterior inference, Eq. 13 is difficult to obtain. Fortunately, this is a scale mixture of a Gaussian with respect to a gamma density [48]. This scale mixture is intrinsically a Student t distribution, and the derivation is following:

$$\begin{aligned} p(y \mid \boldsymbol{\tau_m}) &= \int_{\boldsymbol{\theta_m}} p\left(y \mid \boldsymbol{\theta_m}\right) p(\boldsymbol{\theta_m} \mid \boldsymbol{\tau_m}) \mathrm{d}\boldsymbol{\theta_m} \\ &= \int_{\sigma_m^2=0}^{\infty} \int_{\mu_m=-\infty}^{\infty} p\left(y \mid \mu_m, \sigma_m^2\right) p\left(\mu_m, \sigma_m^2 \mid \boldsymbol{\tau_m}\right) \mathrm{d}\mu_m \mathrm{d}\sigma_m^2 \\ &= \int_{\sigma_m^2=0}^{\infty} \int_{\mu_m=-\infty}^{\infty} p\left(y \mid \mu_m, \sigma_m^2\right) p\left(\mu_m, \sigma_m^2 \mid \delta_m, \gamma_m, \alpha_m, \beta_m\right) \mathrm{d}\mu_m \mathrm{d}\sigma_m^2 \\ &= \int_{\sigma_m^2=0}^{\infty} \int_{\mu_m=-\infty}^{\infty} \left[\sqrt{\frac{1}{2\pi\sigma_m^2}} \exp\left\{-\frac{(y-\mu_m)^2}{2\sigma_m^2}\right\}\right] \left[\frac{\beta_m^{\alpha_m}}{\gamma_m(\alpha_m)} \frac{\sqrt{\gamma_m}}{\sqrt{2\pi\sigma_m^2}}\right. \\ &\qquad \left(\frac{1}{\sigma_m^2}\right)^{\alpha_m+1} \exp\left\{-\frac{2\beta_m + \gamma_m(\delta_m-\mu_m)^2}{2\sigma_m^2}\right\} \mathrm{d}\mu_m \mathrm{d}\sigma_m^2 \\ &= \int_{\sigma_m^2=0}^{\infty} \frac{\beta_m^{\alpha_m}\sigma_m^{-3-2\alpha_m}}{\sqrt{2\pi}\sqrt{1+1/\gamma_m}\Gamma(\alpha_m)} \exp\left\{-\frac{2\beta_m + \frac{\gamma_m(y-\delta_m)^2}{1+\gamma_m}}{2\sigma_m^2}\right\} \mathrm{d}\sigma_m^2 \\ &= \frac{\Gamma(1/2+\alpha_m)}{\Gamma(\alpha_m)} \sqrt{\frac{\gamma_m}{\pi}} (2\beta_m(1+\gamma_m))^{\alpha_m} \left(\gamma_m\left(y-\delta_m\right)^2 + 2\beta_m(1+\gamma_m)\right)^{-\left(\frac{1}{2}+\alpha_m\right)}. \end{aligned}$$

(14)

Accordingly we have:

$$p\left(y \mid \boldsymbol{\tau_m}\right) = \mathrm{St}\left(y; \delta_m, \frac{\beta_m(1+\gamma_m)}{\gamma_m\alpha_m}, 2\alpha_m\right).$$

(15)

For the $m$-th modality, the negative log likelihood loss is defined as Eq. 5. Maximizing the likelihood function by using the standard parameterization for Student t distribution makes our model fit the data. In order to regularize the total evidence, we minimize the evidence of incorrect predictions by adding an incorrect evidence penalty scaled on the error of the predictions. A famous interpretation of the parameters of this kind of conjugate prior distributions is in terms of "virtual observations" [48]. For a sample, the mean of a NIG distribution can be interpreted as being estimated from $\gamma_m$ virtual observations, and the variance can be interpreted as being estimated from $2\alpha_m$ virtual observations with the sum of squared deviations $2\beta_m$ [57], thus, the total evidence can be denoted as the sum of virtual observations: $\Phi_m = \gamma_m + 2\alpha_m$, and the evidence penalty is defined as:

$$\mathcal{L}_m^{\mathrm{R}}(\mathbf{w_m}) = |y - \delta_m| \cdot (\gamma_m + 2\alpha_m).$$

(16)

The total loss for each modality, $\mathcal{L}_m(\mathbf{w_m})$ (Eq. 6), consists of the two terms for maximizing the likelihood function and regularizing evidence.

## B    Experimental details

For experiment on synthetic data, 800 data points are sampled from the range $-4 \le x \le 4$ for training the model, and we present the test data in the region $-7 \le x \le 7$. The model consists of 100

neurons with 4 hidden layers and is trained with the Adam optimizer: $learning\_rate = 5e - 3$ for 60 iterations, $batch\_size = 128$, and the coefficient $\lambda = 0.6$.

For physical and medical tasks, the model consists of 6 hidden layers, trained with the Adam optimizer: $learning\_rate = 1e - 3$ for 400 iterations, and the coefficient $\lambda = 0.05$. Then we impose different degrees of Gaussian noise (i.e., zero mean and varying variance $0.1/0.5$) in different ways to half of the test samples considered as OOD samples, and distinguish them by aleatoric and epistemic uncertainties. Our model still achieves relatively promising performance compared with EVD with concatenating the original features, validating the advantages of estimating the uncertainty of the proposed MoNIG. For human multimodal sentiment analysis, the experimental settings of all the different methods are consistent, with 32 samples for each batch, trained with the Adam optimizer: $learning\_rate = 1e - 3$ for 400 iterations, and the coefficient $\lambda = 0.1$.

**Baseline.** We compare our model with existing approaches including Gaussian (GS) in Eq. 2 and Evidential (EVD) [31], where both the two methods are applied by concatenating features from multiple modalities in two different ways, i.e., early fusion and intermediate fusion. For early fusion, we simply concatenate preprocessed features (data level) as a new representation, while for intermediate fusion, we train each modality with several individual neural layers, then concatenate the last layer of each modality (intermediate level) as a new representation. The new representation is then used as multimodal representation input for prediction.

**Comparison with different decision fusion.** We compare different decision fusion (Fig. 1(b)) strategies: average, weighted average with AU, and weighted average with EU. "Average" indicates the average of multiple results from multiple modalities, and "weighted average with AU" indicates the final prediction is the weighted average of the results from multiple modalities, where the weights are averaged aleatoric uncertainties of every modalities. "Weighted average with EU" indicates the weights are epistemic uncertainties.

**Effectiveness of uncertainty estimation.** We define a rank-based criterion to measure the consistency between uncertainty and prediction error. All test samples are used to compare the rank relationship between uncertainty and RMSE in pairs, where the total number of pairs of test samples is $\text{Num}_{all}$ (i.e., if there are $N$ test samples, $\text{Num}_{all} = \frac{N(N-1)}{2}$). The numbers in the Table 9 indicates the proportion of samples that RMSE and uncertainty are in the opposite order. A smaller value implies a better uncertainty estimation.

## C   Experiments on the rationality and effectiveness of uncertainty estimation

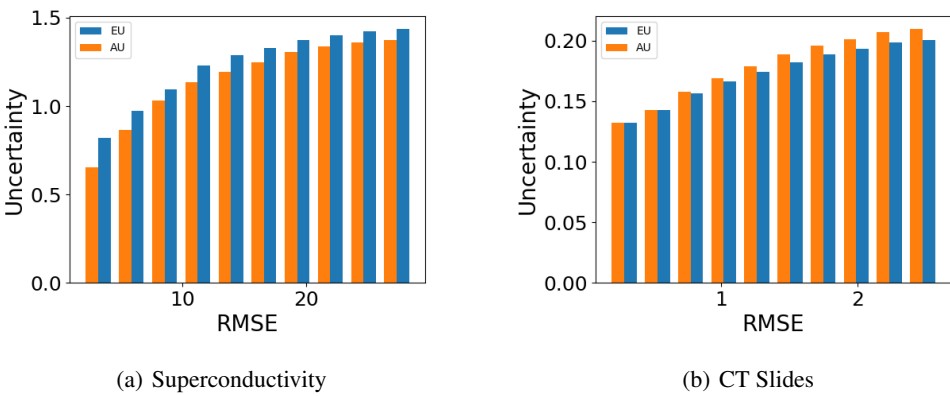

(a) Superconductivity                     (b) CT Slides

Figure 5: Relationship between the uncertainty and RMSE.

We evaluate the aleatoric and epistemic uncertainties of each sample on test data. According to Fig. 5, we can find that both aleatoric uncertainty and epistemic uncertainty are non-decreasing when the RMSE becomes larger, which is quite important to evaluate the trustworthiness for regression.

We show the modality-specific uncertainty estimation on 3-modality data (Fig. 6), which also demonstrates the effectiveness of our model in uncertainty estimation.

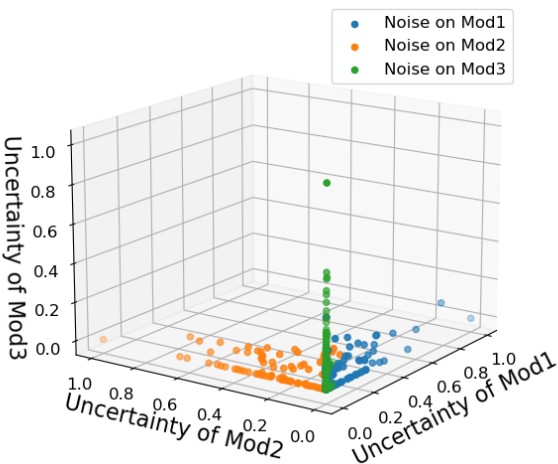

Figure 6: Perception of noisy modality on 3-modality data.

Overall, the experimental results in estimating modality-specific uncertainty as shown in Fig. 6 and Fig. 7(a) clearly show that our method achieves promising performance in estimating modality-specific uncertainty. The modality-specific uncertainty can dynamically guide the integration for different samples, while the counterpart EVD cannot be aware of the different quality of modalities. We also intuitively show why our method makes improvements in (Fig. 7(b)).

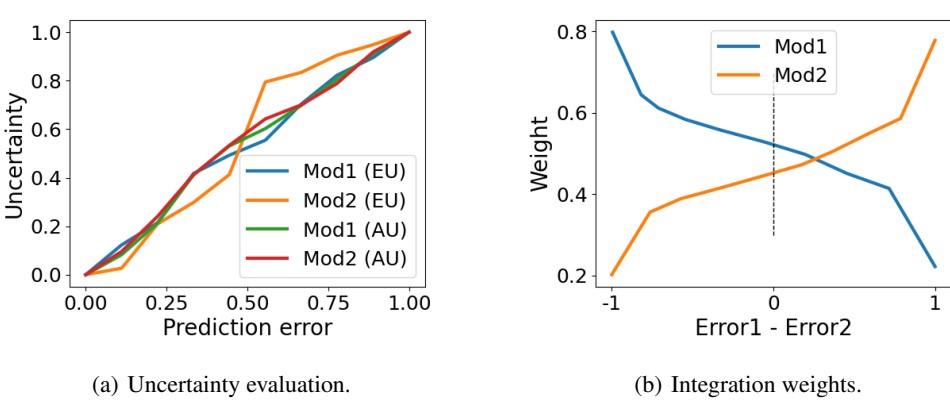

(a) Uncertainty evaluation.

(b) Integration weights.

Figure 7: Left: The relationship between the modality-specific uncertainty and prediction error, which validates the rationality of modality-specific uncertainty. Right: The relationship between the weight of each modality to the final prediction and the absolute error comparison of two modalities' predictions. "Error1 - Error2" indicates the normalized difference between the Mod1's prediction error and Mod2's prediction error, i.e., "-1" indicates Mod1 can give a much more accurate prediction than Mod2, and vice versa. We can find that the modality with a smaller error has a larger weight for the final prediction and and vice versa, which potentially reduces the impact from noisy modality.

# D Analysis of the training time and space complexity

We report the training time on the sentiment analysis task (Table 10). It is observed that although the NIG summation operation is introduced for multimodal integration our method has the same level of computational complexity.

Table 10: Training time on MOSI. (Platform: RTX 2080)

| Method | GS | EVD | Ours |
|---|---|---|---|
| Time (s) | 2819.85 | 2869.59 | 2981.52 |

# E  Proof of the proposition

The commutativity and associativity of summation in Def. 4.1 can be proved as follows:

**Commutativity:**

$$NIG_{\delta_1}\left(\delta_1, \gamma_1, \alpha_1, \beta_1\right) \oplus NIG_{\delta_2}\left(\delta_2, \gamma_2, \alpha_2, \beta_2\right)$$
$$= (\gamma_1 + \gamma_2)^{-1}(\gamma_1\delta_1 + \gamma_2\delta_2)$$
$$= (\gamma_2 + \gamma_1)^{-1}(\gamma_2\delta_2 + \gamma_1\delta_1)$$
$$= NIG_{\delta_2}\left(\delta_2, \gamma_2, \alpha_2, \beta_2\right) \oplus NIG_{\delta_1}\left(\delta_1, \gamma_1, \alpha_1, \beta_1\right)$$

$$NIG_{\gamma_1}\left(\delta_1, \gamma_1, \alpha_1, \beta_1\right) \oplus NIG_{\gamma_2}\left(\delta_2, \gamma_2, \alpha_2, \beta_2\right)$$
$$= \gamma_1 + \gamma_2$$
$$= \gamma_2 + \gamma_1$$
$$= NIG_{\gamma_2}\left(\delta_2, \gamma_2, \alpha_2, \beta_2\right) \oplus NIG_{\gamma_1}\left(\delta_1, \gamma_1, \alpha_1, \beta_1\right)$$

$$NIG_{\alpha_1}\left(\delta_1, \gamma_1, \alpha_1, \beta_1\right) \oplus NIG_{\alpha_2}\left(\delta_2, \gamma_2, \alpha_2, \beta_2\right)$$
$$= \alpha_1 + \alpha_2$$
$$= \alpha_2 + \alpha_1$$
$$= NIG_{\alpha_2}\left(\delta_2, \gamma_2, \alpha_2, \beta_2\right) \oplus NIG_{\alpha_1}\left(\delta_1, \gamma_1, \alpha_1, \beta_1\right)$$

$$NIG_{\beta_1}\left(\delta_1, \gamma_1, \alpha_1, \beta_1\right) \oplus NIG_{\beta_2}\left(\delta_2, \gamma_2, \alpha_2, \beta_2\right)$$
$$= \beta_1 + \beta_2 + \frac{1}{2}\gamma_1(\delta_1 - \delta)^2 + \frac{1}{2}\gamma_2(\delta_2 - \delta)^2$$
$$= \beta_2 + \beta_1 + \frac{1}{2}\gamma_2(\delta_2 - \delta)^2 + \frac{1}{2}\gamma_1(\delta_1 - \delta)^2$$
$$= NIG_{\beta_2}\left(\delta_2, \gamma_2, \alpha_2, \beta_2\right) \oplus NIG_{\beta_1}\left(\delta_1, \gamma_1, \alpha_1, \beta_1\right)$$

**Associativity:**

$$NIG_{\delta_1}\left(\delta_1, \gamma_1, \alpha_1, \beta_1\right) \oplus NIG_{\delta_2}\left(\delta_2, \gamma_2, \alpha_2, \beta_2\right) \oplus NIG_{\delta_3}\left(\delta_3, \gamma_3, \alpha_3, \beta_3\right)$$
$$=((\gamma_1 + \gamma_2) + \gamma_3)^{-1}((\gamma_1 + \gamma_2)(\gamma_1 + \gamma_2)^{-1}(\gamma_1\delta_1 + \gamma_2\delta_2) + \gamma_3\delta_3)$$
$$=(\gamma_1 + (\gamma_2 + \gamma_3))^{-1}(\gamma_1\delta_1 + (\gamma_2 + \gamma_3)(\gamma_2 + \gamma_3)^{-1}(\gamma_2\delta_2 + \gamma_3\delta_3))$$
$$=NIG_{\delta_1}\left(\delta_1, \gamma_1, \alpha_1, \beta_1\right) \oplus [NIG_{\delta_2}\left(\delta_2, \gamma_2, \alpha_2, \beta_2\right) \oplus NIG_{\delta_3}\left(\delta_3, \gamma_3, \alpha_3, \beta_3\right)]$$

$$NIG_{\gamma_1}\left(\delta_1, \gamma_1, \alpha_1, \beta_1\right) \oplus NIG_{\gamma_2}\left(\delta_2, \gamma_2, \alpha_2, \beta_2\right) \oplus NIG_{\gamma_3}\left(\delta_3, \gamma_3, \alpha_3, \beta_3\right)$$
$$=(\gamma_1 + \gamma_2) + \gamma_3$$
$$=\gamma_1 + (\gamma_2 + \gamma_3)$$
$$=NIG_{\gamma_1}\left(\delta_1, \gamma_1, \alpha_1, \beta_1\right) \oplus [NIG_{\gamma_2}\left(\delta_2, \gamma_2, \alpha_2, \beta_2\right) \oplus NIG_{\gamma_3}\left(\delta_3, \gamma_3, \alpha_3, \beta_3\right)]$$

$$NIG_{\alpha_1}\left(\delta_1, \gamma_1, \alpha_1, \beta_1\right) \oplus NIG_{\alpha_2}\left(\delta_2, \gamma_2, \alpha_2, \beta_2\right) \oplus NIG_{\alpha_3}\left(\delta_3, \gamma_3, \alpha_3, \beta_3\right)$$
$$=(\alpha_1 + \alpha_2) + \alpha_3$$
$$=\alpha_1 + (\alpha_2 + \alpha_3)$$
$$=NIG_{\alpha_1}\left(\delta_1, \gamma_1, \alpha_1, \beta_1\right) \oplus [NIG_{\alpha_2}\left(\delta_2, \gamma_2, \alpha_2, \beta_2\right) \oplus NIG_{\alpha_3}\left(\delta_3, \gamma_3, \alpha_3, \beta_3\right)]$$

$$NIG_{\beta_1}(\delta_1, \gamma_1, \alpha_1, \beta_1) \oplus NIG_{\beta_2}(\delta_2, \gamma_2, \alpha_2, \beta_2) \oplus NIG_{\beta_3}(\delta_3, \gamma_3, \alpha_3, \beta_3)$$

$$= [\beta_1 + \beta_2 + \frac{1}{2}\gamma_1(\delta_1 - (\gamma_1 + \gamma_2)^{-1}(\gamma_1\delta_1 + \gamma_2\delta_2))^2$$

$$+ \frac{1}{2}\gamma_2(\delta_2 - (\gamma_1 + \gamma_2)^{-1}(\gamma_1\delta_1 + \gamma_2\delta_2))^2]$$

$$+ \beta_3 + \frac{1}{2}(\gamma_1 + \gamma_2)((\gamma_1 + \gamma_2)^{-1}(\gamma_1\delta_1 + \gamma_2\delta_2) - \delta)^2$$

$$+ \frac{1}{2}\gamma_3(\delta_3 - \delta)^2$$

$$= \frac{1}{2}(\gamma_2 + \gamma_3)((\gamma_2 + \gamma_3)^{-1}(\gamma_2\delta_2 + \gamma_3\delta_3) - \delta)^2$$

$$+ \beta_1 + \frac{1}{2}\gamma_1(\delta_1 - \delta)^2 + [(\beta_2 + \beta_3$$

$$+ \frac{1}{2}\gamma_2(\delta_2 - (\gamma_2 + \gamma_3)^{-1}(\gamma_2\delta_2 + \gamma_3\delta_3))^2$$

$$+ \frac{1}{2}\gamma_3(\delta_3 - (\gamma_2 + \gamma_3)^{-1}(\gamma_2\delta_2 + \gamma_3\delta_3))^2]$$

$$= NIG_{\beta_1}(\delta_1, \gamma_1, \alpha_1, \beta_1) \oplus [NIG_{\beta_2}(\delta_2, \gamma_2, \alpha_2, \beta_2) \oplus NIG_{\beta_3}(\delta_3, \gamma_3, \alpha_3, \beta_3)]$$