# OpenReview forum: "Trustworthy Multimodal Regression with Mixture of Normal-inverse Gamma Distributions"
_NeurIPS.cc/2021/Conference — NeurIPS 2021 Poster_

### Official Review · Reviewer_BXmu · 2021-07-11

**Rating:** 8
**Confidence:** 3

**Summary:**

This paper proposes a Mixture of Normal-Inverse Gamma distributions algorithm to estimate uncertainty in principle for adaptive integration of different modalities.

This research problem proposed in this paper is very important, and the author proposed some techniques and conducted some experiments.

**Limitations And Societal Impact:**

This paper does not clarify the limitations of the proposed method.

In fact, the method proposed by the author has some assumptions about the distribution of the data. The rationality of the assumptions needs to be explained.

**Main Review:**

This paper is innovative to some extent. In fact, it is difficult to measure the uncertainty of combining the information from different modalities.

The writing of the paper needs further polishing. Some details of the paper are not detailed. For example, Figure 1 and Figure 2 are both confusing.

**Time Spent Reviewing:**

1.5

---

> ### Author Response · Authors · 2021-08-10
> **Response to Reviewer BXmu**
>
> Thanks for your comments.
>
> Q1: “It is difficult to measure the uncertainty of combining the information from different modalities.”
>
> R1: Yes, estimating uncertainty of combining the information from different modalities is very challenging. In this work, we estimate the modality-specific uncertainty by NIG distribution, and then we fuse multiple NIG distributions into a uniform NIG through MoNIG which is a well-established mathematical technique. The experimental results on both synthetic and real-world data also demonstrate the effectiveness in uncertainty estimation.
>
> Q2: “Figure 1 and Figure 2 are both confusing.”
>
> R2: (1) We compare three different fusion strategies in the Fig. 1. Fig. 1(a) demonstrates the feature fusion, which jointly learns unified representations and based on which conducts regression. We compared ours with it as baselines in the Table 1 and Table 2. Fig. 1 (b) indicates the decision fusion, which trains one regression model for each modality. The final prediction is regarded as a function of the predictions of all modalities. We show details in the supplement section B (line 532). For the Fig. 1(c), we show the framework of the proposed MoNIG. Fig.1 (d) is an instance of MoNIG on the multimodal sentiment analysis task. We will further clarify this by adding description in the caption.
> (2) We show the regression curves for each modality and the fused ones (with MoNIG) in the Fig. 2(a). Fig. 2(b) demonstrates the estimated aleatoric uncertainty. Fig.2 (c) demonstrate the estimated epistemic uncertainty. We will clarify this by adding description in the caption.
>
> Q3: This paper does not clarify the limitations of the proposed method.
>
> R3: Since there is no closed form solution for mixture of NIG distributions, we can only approximate it with the existing mathematical methods in practice. More accurate even exact fusion techniques need to be explored in the future.
>
> Q4: The assumptions about the distribution of the data.
>
> R4: Actually, we assume that the target $y$ is drawn from a Gaussian distribution, which is a conjugate prior distribution of NIG distribution. what’s more, Gaussian distribution is widely used because of its generality. Experiments also verify that relatively promising performance can be achieved under this assumption.

---

> > ### Comment · Reviewer_BXmu · 2021-08-12
> > **Response to Paper352**
> >
> > Thank you for your reply! You resolved my doubts.
> >
> > The problem of determining the uncertainty of combining the information from different modalities is indeed a very important problem in multimodal learning.
> >
> > As the first related work in the academic community, the NIG distribution you proposed for estimating the uncertainty in multimodal learning is good both theoretically and experimentally.
> >
> > It would be better for the author to show experimental results more clearly.

---

> > > ### Author Response · Authors · 2021-08-30
> > > **Response to Reviewer BXmu**
> > >
> > > Thanks for the positive comments and valuable suggestions. We will clarify the experimental results and updated our paper according to the responses.

---

### Official Review · Reviewer_8nwK · 2021-07-15

**Rating:** 8
**Confidence:** 4

**Summary:**

This paper focuses on an important problem in multimodal learning - current algorithms ignore the confidence of integration and prediction.

**Limitations And Societal Impact:**

Minors：
-It is an elegant model for trustworthy multimodal regression. Can the model be applied or extended for classification task? It will be better if the authors could provide related discussion.
- The proposed method introduces uncertainty estimation, so what about the computational complexity of the proposed model compared with existing multimodal algorithms?

**Main Review:**

This paper focuses on an important problem in multimodal learning - current algorithms ignore the confidence of integration and prediction. The authors smartly use Mixture of Normal Inverse Gamma distributions (MoNIG) to estimate uncertainty for multimodal learning. The model estimates uncertainty for each single modality and the final prediction as well. Experiments on both synthetic and real datasets well support the assumptions.

Strengths:
- Trustworthy multimodal learning is important and interesting, and the motivation of the paper is very clear and strong.
-The introduced pseudo modality is interesting and necessary, which enhances the sufficient multimodal interactions beyond explainability.
-Both the detailed theoretical analysis and experimental results strongly support the motivation. Especially, the uncertainty estimation and ablation study provide strong evidence to show which parts contribute to the improvements.
-The writing and organization are good making the paper quite easy to follow.

Minors：
-It is an elegant model for trustworthy multimodal regression. Can the model be applied or extended for classification task? It will be better if the authors could provide related discussion.
- The proposed method introduces uncertainty estimation, so what about the computational complexity of the proposed model compared with existing multimodal algorithms?


**Time Spent Reviewing:**

0.5 hour

---

> ### Author Response · Authors · 2021-08-10
> **Response to Reviewer 8nwK**
>
> We appreciate for the identification of our novelty and the positive comments. We believe the following response can address the concerns.
>
> (1) The proposed method is designed for regression; however, it is also suitable for rank-based classification, e.g., multimodal sentiment analysis. We will consider extending this work to more general cases in the future.
>
> (2) We have reported the training time on the sentiment analysis task in the supplement materials, which is observed that our method has the same order of computational complexity compared to existing models. As for spatial complexity, suppose that the spatial complexity for each modality is "$C$", the spatial complexity of our method is "$KC$", where "$K$" is the number of modalities.

---

### Official Review · Reviewer_4yAM · 2021-07-16

**Rating:** 6
**Confidence:** 4

**Summary:**

This paper tackles a shortcoming of existing multimodal learning methods that mainly focus on the integration of the information from different sources for prediction but ignore the confidence of prediction. The authors propose a Mixture of NormalInverse Gamma distributions (MoNIG) algorithm which estimates uncertainty for multimodal learning. Their model estimates uncertainty for each modality and also in corrupted modality settings.

**Ethical Concerns:**

There are no ethical concerns as far as I can tell.

**Limitations And Societal Impact:**

I think the authors can say a bit more about the societal impact of multimodal models, including their applications in robotics, healthcare, education etc.

**Main Review:**

Strengths:
1. The problem tackled is an important one and an important shortcoming in current work.
2. The proposed method is a nice extension of uncertainty estimation for single modality regression problems using Normal-inverse Gamma Distributions into one for multimodal problems using a mixture of distributions.
3. Experimental results are quite comprehensive on 3 sets of tasks spanning a few multimodal datasets.
4. Experimental results on prediction tasks are strong and show some improvement over current methods.
5. Experimental results also show that the model can work well in the presence of noisy modalities.

Weaknesses:
1. There can be more analysis of when the proposed method fails and other potential shortcomings of their approach.
2. There should be more analysis on what types of multimodal inputs and modalities the model estimates uncertainty for, including visualizations and qualitative examples.
3. The paper can benefit from some additional analysis and discussion of the additional training time and space complexity resulting from their model.
4. There should be some reference to recent work in learning robust multimodal representations [1,2].
5. The model currently handles multimodal interactions using a very simple late-fusion style mixture of distributions - how can the method be extended to more fine-grained multimodal representation learning methods such as those in Table 5 (e.g. RMFN, MFM etc.)
6. There is not enough evaluation to show that the model has accurately estimated uncertainty apart from some visualizations. How do you evaluate whether your model has estimated uncertainty correctly? Are there any metrics for judging this? I think this might be a good reference: https://github.com/uncertainty-toolbox/uncertainty-toolbox

[1] Liang et al., Learning Representations from Imperfect Time Series Data via Tensor Rank Regularization. ACL 2019

[2] Lee et al., Detect, Reject, Correct: Crossmodal Compensation of Corrupted Sensors. 2020

Not a weakness, but a general suggestion: The authors can take a look at https://arxiv.org/abs/2107.07502 for a wide suite of multimodal methods and datasets that they can test their method on.

**Time Spent Reviewing:**

2 hours

---

> ### Author Response · Authors · 2021-08-10
> **Response to Reviewer 4yAM**
>
> Q1: Potential shortcomings of our approach.
>
> R1: (1) Consider that it is risky for the cases that each modality is insufficient to make trustworthy decision due to partial observation, we have introduced the feature-level fusion producing a pseudo modality, which makes full use of the complementary information. Accordingly, it makes the computational complexity higher. (2) Since there is no closed form solution for mixture of NIG distributions, we can only approximate it with the existing mathematical methods in practice. More accurate even exact fusion techniques need to be explored in the future.
>
> Q2: “There should be more analysis on what types of multimodal inputs and modalities the model estimates uncertainty for, including visualizations and qualitative examples.”
>
> R2: Our method is suitable for both preprocessed features and raw data. In our experiments, we employ three real datasets: (1) Superconductivity consists of raw data, such as “mean atomic mass” and “zinc atom content in chemical formula”. (2) CT slices consists of feature vectors extracted from CT image by doctor, such as bone structures. (3) Multimodal sentiment analysis is preprocessed data. For all datasets, we provide download links in footnotes.
>
> Q3: Training time and space complexity. The paper can benefit from some additional analysis and discussion of the additional training time and space complexity resulting from their model.
>
> R3: In practice, we have reported the training time on the sentiment analysis task in the supplement materials (Table 10), which implies that our method has the same level of computational complexity with traditional models. In principle, our model is a multimodal decision fusion algorithm, so the space complexity and time complexity are the same order as backbone models.
> As for spatial complexity, suppose that the spatial complexity for each modality is "$C$", the spatial complexity of our method is "$KC$", where "$K$" is the number of modalities.
>
> Q4: Some reference to recent work in learning robust multimodal representations.
>
> R4: Thanks for suggestion. These two methods achieve SOTA under imperfect multimodal data which are inspiring, and we will add them as references.
>
> Q5: The model currently handles multimodal interactions using a very simple late-fusion style mixture of distributions - how can the method be extended to more fine-grained multimodal representation learning methods such as those in Table 5 (e.g., RMFN, MFM etc.)
>
> R5: Thanks for the suggestion. Our work proposes two kinds of fusion styles, one is late-fusion without fine-grained multimodal representation learning, and the other introduces the feature-level fusion producing an additional pseudo modality, which makes full use of the complementary information. The pseudo modality could be generated by using off-the-shelf fine-grained multimodal representation learning methods. So, our method explores both fine-grained multimodal representation learning and final decision fusion (late fusion). We will emphasize this in the revision.
>
> Q6: How do you evaluate whether your model has estimated uncertainty correctly?
>
> R6: Thank you for suggestion. We would like to clarify that exiting evaluation metrics are typically designed to quantify the confidence intervals, however, our method estimates predictive uncertainty by aleatoric uncertainty and epistemic uncertainty. The meanings of aleatoric uncertainty and epistemic uncertainty are different from confidence intervals, so those metrics are not suitable for them. Even though, to judge whether our model has estimated uncertainty reasonably, we evaluate them by a rank-based criterion, as shown in Table 9, and the detailed description could be found in section 5.6 and supplementary material.
>
>
> Q7: “I think the authors can say a bit more about the societal impact of multimodal models.”
>
> R7: It is important and interesting to apply the trustworthy multimodal regression for more real-world applications in the future, where the uncertainty estimation is even more important than the accuracy in safety-critical applications, e.g., robotics, medical diagnosis tasks.

---

> > ### Comment · Reviewer_4yAM · 2021-08-28
> > **thank you for your response**
> >
> > Thank you authors for your response - I remain positive about the paper. However, I do hope that the authors can add my suggestions to the paper in more detail, such as “more analysis on what types of multimodal inputs and modalities the model estimates uncertainty for, including visualizations and qualitative examples.”, and "how do you evaluate whether your model has estimated uncertainty correctly?" Otherwise I think it is a good paper.

---

> > > ### Author Response · Authors · 2021-08-30
> > > **Thanks for the positive comments - will update according to suggestions**
> > >
> > > We appreciate the positive comments and valuable suggestions, and according to the suggestions we will add the related explanations and experiments (some are included in the rebuttals)  in the updated version.
> > >
> > > For the types of multimodal inputs and modalities, we would like to clarify the following aspects in the updated version. First, the detailed description (including the descriptions in rebuttal) of modalities used will be added. Second, we will further clarify the noised inputs, e.g., some heavily corrupted modalities. Finally, the visualization experiments have been conducted according to the reviewer's suggestions, and will be added in the updated version.
> > >
> > > For the uncertainty evaluation in multimodal learning, we designed a new (rank-based) criterion for uncertainty evaluation, and the new evaluation results (including the results in supplement) will be calrified in the updated version.
> > >
> > > Thanks.

---

### Official Review · Reviewer_vgoM · 2021-07-17

**Rating:** 5
**Confidence:** 3

**Summary:**

The problem of multi-modal regression is considered in this paper and a method called Mixture of Normal-Inverse Gamma (MoNIG) is proposed. In the proposed method, an approximation is adopted to combine predictive normal-inverse gamma distributions from different modalities into one normal-inverse gamma for tractable uncertainty calculation.

**Limitations And Societal Impact:**

The authors have not discussed the limitations and societal impact of their work.

**Main Review:**

The paper aims to develop a new method for trustworthy multi-modal regression which is an important problem. The performance is compared with Gaussian and Evidential Regression with concatenated features/hidden layers on two datasets and on a multimodal sentiment analysis with several baselines. The results show improvements over the baselines.

Concerns:

The methodology of the paper is a straightforward combination of existing techniques, Deep Evidential Regression, and the summation operation for NIG distributions. Thus, the methodological contribution of the paper is not very significant.

The NIG summation operation is originally introduced for Bayesian linear regression on large datasets, which is exact there, but it is not clear in the paper how valid the approximation is for the paper’s setup.

For the experiments, simple baselines which are fitting GS or EVD to each modality and then averaging them, simple or weighted, are not considered.

Results in Table 7 suggest that the performance of the method still deteriorates even though noise is only added to one modality. So, the discussion in the paper regarding robustness to modality corruption can be adjusted accordingly.

None of the experiments contain error bars, so it is unclear how significant are some of the results, especially those with closer numbers.

The paper has some issues in terms of clarity. The provided details are not enough to reproduce the results. In Tables 5 and 6 the baselines do not have references. The details are not provided for the ablation studies.

Additional analysis like Figure 4 can also be done for the baselines for comparison.


--Update--

Thanks to the authors for their responses. The authors have clarified what Table 8 shows (I assume the weights are per sample) and provided some error bars which address some of my concerns. Hence, I have increased the score by 1 point.

My major concern about the work remains up to some extent. While the insight of combining existing techniques for multimodal regression is new, there does not appear to be any significant technical challenges involved to make them work here. Additionally, in my view, the paper has room for improvement and in its current form does not properly answer why the uncertainties from applying the NIG summation in the paper’s setup are valid and reliable.

A few other suggestions to strengthen the experiments:

-Calibration curves of the estimated uncertainties can be included

-Modality and “pseudo-modality” specific uncertainties can be compared systematically with and without the NIG summation. Currently, there is no comparison in this regard.



**Time Spent Reviewing:**

2-3

---

> ### Author Response · Authors · 2021-08-10
> **Response to Reviewer vgoM**
>
> Thank you for carefully reviewing our work and valuable suggestions. We believe the following point-to-point response can address all the concerns:
>
> Q1: “Contribution”.
>
> R1: (1) To the best of our knowledge, our work is the first method toward trustworthy multimodal regression, as Reviewer 2 and Reviewer 3 point out: “This paper focuses on an important problem in multimodal learning - current algorithms ignore the confidence of integration and prediction”, “The problem tackled is an important one and an important shortcoming in current work”. A trustworthy prediction is important, especially when the systems are deployed in safety-critical domains, such as automated driving, healthcare, etc. (2) The summation operation for NIG distributions is a simple but effective mathematical formulation. It is the first time to be introduced into the multimodal learning, and the experiment demonstrates that it is suitable for multimodal regression. (3) We have conducted plenty of experiments on both synthetic and real datasets which prove the rationality and effectiveness of the proposed method. (4) Moreover, we also introduce a pseudo modality for the feature level fusion, which makes full use of the complementary information.
>
> Q2: “How valid the approximation is for the paper’s setup?”
>
> R2: The NIG summation has been proved that it is reasonable to fuse two NIG distributions in terms of pseudo-observation interpretation (please refer to reference [41]). In our paper, we map each modality to a NIG distribution about the target or output through neural network firstly, and the information of different modalities are transformed into a same space (i.e., output space). Then, similar to distribution fusion in Bayesian linear regression [41], we fuse the NIG distributions obtained from different modalities. Different to [41] merging the NIG distributions from different data, we fuse NIG distributions from different modalities. Our additive operation is essentially the same as data distribution fusion. In other words, we firstly introduce the mathematical formulation to multimodal learning.
>
> Q3: Experiments baselines. For the experiments, simple baselines which are fitting GS or EVD to each modality and then averaging them, simple or weighted, are not considered.
>
> R3: GS and EVD are typical and SOTA methods which can estimate uncertainty. In experiments, we actually provide average, concatenating the features from hidden layer or the original features, and weighted average as shown in Table 8.
>
>
> Q4: “Results in Table 7 suggest that the performance of the method still deteriorates even though noise is only added to one modality. So, the discussion in the paper regarding robustness to modality corruption can be adjusted accordingly.”
>
> R4: We will add discussion about the modality corruption. Specifically, it's very challenging to maintain the high performance under the corrupted modality cases. Fortunately, although all methods are affected by corruption, our method still outperforms other baselines as shown in Table 7. Meanwhile, our model enhances the trustworthiness via modalities uncertainty and final decision uncertainty, which allows safe decision.
>
> Q5: Error bars.
>
> R5: We did run experiments multiple times, but considering that the limited space and the result is relatively stable, we only show the average value in some tables. According to the suggestion of the reviewer 1, we have added error bars in the revision.
>
> Table 1
> $$
> \begin{array}{cccccc}
> \hline \text { Modl } & \text { Mod2 } & \text { GS (IF/DF) } & \text { EVD (IF/DF) } & \text { Ours } & \text { Ours (Pseudo) } \\\\
> \hline 1 2 . 9 3 ( 0 . 3 1 ) & 12.82(0.42) & 19.93(0.34) / 13.78(0.25) & 14.18(0.46) / 12.79(0.29) & 12.19(0.37) & \mathbf{11.70(0.26)} \\\\
> \hline
> \end{array}
> $$
>
> Table 3
> $$
> \begin{array}{cccccc}
> \hline \text { Mod1 } & \text { Mod2 } & \text { GS (IF/DF) } & \text { EVD (IF/DF) } & \text { Ours } & \text { Ours (Pseudo) } \\\\
> \hline 1 . 6 7 ( 0 . 1 2 ) & 4.49(0.27) & 1.64(0.11) / 1.70(0.06) & 2.97(0.18) / 3.27(0.07) & 0.91(0.08) & \mathbf{0.79(0.07)} \\\\
> \hline
> \end{array}
> $$
>
> Q6: Some baselines do not have references.
>
> R6: We will add these related references.
>
> Q7: “Additional analysis like Figure 4 can also be done for the baselines for comparison.”
>
> R7: (1) For Figure 4, we mainly aim to investigate why our method achieves better fusion performance based on uncertainty. Therefore, we do not show other methods; (2) Other methods cannot estimate both the modality-specific uncertainty and the global uncertainty under the unified (optimizable) objective; (3) Even so, we still conduct experiments of other methods as suggested by Reviewer 1, which will be added in the supplement. As shown in the figures, due to little interaction of uncertainty between different modalities, the compared methods (Gaussian and EVD) cannot well identify the noisy modality compared to ours.

---

> > ### Comment · Area_Chair_bwRx · 2021-08-30
> > **Could you please clarify this point?**
> >
> > Dear authors,
> >
> > Thank you for your response. I went through your rebuttal to the reviewer vgoM, however, I also feel this point has not been addressed well.
> > So, could you please give us more feedback?
> >
> > > The methodology of the paper is a straightforward combination of existing techniques, Deep Evidential Regression, and the summation operation for NIG distributions.
> >
> >  We do agree that the application to the multimodal regression part is new. Is your method a simple combination of DER and the summation operation for NIG distribution? Or, is there any significant contribution from a technical perspective?
> >
> >  Thanks,
> >
> > AC

---

> > > ### Author Response · Authors · 2021-08-30
> > > **Thanks for the comments, clarification on technical contributions**
> > >
> > > Dear AC and Reviewers:
> > >
> > > Thanks a lot for the insightful suggestions. We would like to clarify the contributions from technique perspectives:
> > >
> > > (1)	The formulation of trustworthy multimodal regression is novel. It provides a stable and explainable model in multimodal regression. This is a new multimodal regression paradigm.
> > >
> > > (2)	In fusion strategy, the late fusion equipped with pseudo modality is technically novel, which simultaneously addresses multimodal interactions and trustworthiness (4.1 Overview of Fusion Strategy). For example, it well addresses the case that each modality is insufficient to make trustworthy decision due to partial observation. This is technically novel in multimodal learning.
> > >
> > > (3)	Although the summation of NIG is introduced in existing work [A], it is a mathematical model for big data as stated in abstract of the paper [A] “We introduce the normal-inverse-gamma summation operator, which combines Bayesian regression results from different data sources and leads to a simple split-and-merge algorithm for big data regressions”. While we provide a full technical framework (including pseudo modality and multi-task objective) for trustworthy multimodal regression. It is not trivial to extend single-modal for multimodal regression by properly using math techniques.
> > >
> > > (4)	We provide the discussion of why the summation of NIG is suitable for trustworthy multimodal regression (Flexibility/Explainability/Trustworthiness/Optimizability), although further analysis is needed.
> > >
> > > Hopefully, AC/Reviewers could note that this is the first attempt in trustworthy multimodal regression, so we mainly focus on a reasonable framework and adopting reasonable techniques for our task. We believe the framework is inspiring and more advanced realizations for the task will be investigated in the future. We will clarify above clarifications in the final version.
> > >
> > > [A] Big Data Bayesian Linear Regression and Variable Selection by Normal-Inverse-Gamma Summation
> > >
> > > Thanks all for the constructive suggestions.

---

### Author Response · Authors · 2021-08-10
**Summary of Response**

Some experiments have been conducted and discussed in the supplement which may reduce some misunderstandings, and a more detailed explanation is given below according to the comments of the reviewers.

---

### Decision · Program_Chairs · 2021-09-27

**Decision:**

Accept (Poster)

**Comment:**

In this paper, the authors propose a new regression method for multi-modal (e.g., image and text) data. More specifically, the authors extend the Deep Evidential Regression method for a multi-modal regression problem. The method depends heavily on the existing methods. However, the formulation is new and the fusion strategy has some merit. Thus, I also vote for an acceptance. For the camera-ready version, I expect authors to revise the paper based on the reviewer's comments.